# Is Serum 25-Hydroxyvitamin D Level Associated with Severity of COVID-19? A Retrospective Study

**DOI:** 10.3390/jcm12175520

**Published:** 2023-08-25

**Authors:** Munachimso Kizito Mbata, Mireille Hunziker, Anja Makhdoomi, Giorgia Lüthi-Corridori, Maria Boesing, Stéphanie Giezendanner, Jürgen Muser, Anne B. Leuppi-Taegtmeyer, Jörg D. Leuppi

**Affiliations:** 1Cantonal Hospital Baselland, University Center of Internal Medicine, Rheinstrasse 26, 4410 Liestal, Switzerland; 2Faculty of Medicine, University of Basel, Klingelbergstrasse 61, 4056 Basel, Switzerland; 3Center for Rehabilitation and Geriatrics, Cantonal Hospital Baselland, Gemeindeholzweg, 4101 Bruderholz, Switzerland; 4Central Laboratories, Cantonal Hospital Baselland, Rheinstrasse 26, 4410 Liestal, Switzerland; 5Hospital Pharmacy, Cantonal Hospital Baselland, Rheinstrasse 26, 4410 Liestal, Switzerland; 6Department of Patient Safety, Medical Directorate, University Hospital Basel, Schanzenstrasse 55, 4056 Basel, Switzerland

**Keywords:** COVID-19, COVID-19 severity, 25-hydroxyvitamin D, vitamin D deficiency, acute respiratory distress syndrome, retrospective study

## Abstract

(1) Background: SARS-COV2 infection has a clinical spectrum ranging from asymptomatic infection to COVID-19 with acute respiratory distress syndrome (ARDS). Although vitamin D deficiency is often found in patients with ARDS, its role in COVID-19 is not clear. The aim of this study was to explore a possible association between serum 25-hydroxyvitamin D levels and the severity of COVID-19 in hospitalised patients. (2) Methods: In this retrospective observational study, we analysed data from 763 patients hospitalised for COVID-19 in 2020 and 2021. Patients were included in the study if serum 25-hydroxyvitamin D was assessed 30 days before or after hospital admission. Vitamin D deficiency was defined as <50 nmol/L (<20 ng/mL). The primary outcome was COVID-19 severity. (3) Results: The overall median serum 25-hydroxyvitamin D level was 54 nmol/L (IQR 35–76); 47% of the patients were vitamin D deficient. Most patients had mild to moderate COVID-19 and no differences were observed between vitamin D deficient and non-deficient patients (81% vs. 84% of patients, respectively *p* = 0.829). (4) Conclusion: No association was found between serum 25-hydroxyvitamin D levels and COVID-19 severity in this large observational study conducted over 2 years of the pandemic.

## 1. Introduction

On 11 March 2020, the World Health Organization (WHO) declared an outbreak of the novel coronavirus 2019 (2019-nCoV) a global pandemic [1]. It originally started in late 2019 in Wuhan, Hubei, China and was renamed severe acute respiratory syndrome coronavirus 2 (SARS-CoV-2) in February 2020 [2,3]. The clinical spectrum of the associated disease, COVID-19, is broad and ranges from asymptomatic and mild upper respiratory infections to potentially lethal acute respiratory distress syndrome (ARDS) [4] and multi-organ failure [5]. Other COVID-19 manifestations, such as uni- or bilateral pulmonary infiltrates in chest imaging like X-ray or computed tomography (CT), can be observed [6].

The mortality of COVID-19 is determined by various factors, including country of residence, patients’ comorbidities, the treatment setting and time of infection [7,8,9]. In Switzerland, the in-hospital mortality was 9.3% [8], while the overall mortality was 1.0% [7] between January 2020 and December 2021. During the two years of the pandemic, patient numbers, symptoms and especially therapeutic guidelines changed rapidly because the virus and its mutations were initially unknown.

Vitamin D is a fat-soluble vitamin, which needs to be introduced into the body by diet or supplementation (in the form of vitamin D_2_ (ergocalciferol) or vitamin D_3_ (cholecalciferol)) or produced in the skin from precursors that are converted by sunlight (specifically via ultraviolet B rays) into vitamin D_3_ [10]. Further, 25-hydroxyvitamin D is the major circulating form of vitamin D; it has a halflife of approximately 2–3 weeks. Additionally, 25-hydroxyvitamin D is a summation of both vitamin D intake and vitamin D that is produced from sun exposure and is the only vitamin D metabolite that is used to determine whether a patient is vitamin D deficient or sufficient.

It has been shown that vitamin D deficiency is associated with various diseases, such as diabetes, viral infections, immune dysregulation, inflammation, cardiovascular disease, cognitive decline and others [11,12,13]. This knowledge has led to studies about vitamin D and COVID-19, which showed contradictory results, especially when differentiating between randomised controlled trials (RCTs) and observational studies [14]. In a meta-analysis of eleven cohort studies and two RCTs, Chen et al. (2021) highlighted no significant association between increased risk of COVID-19 infection or in-hospital death with vitamin D deficiency or insufficiency [15]. However, within the different studies of the meta-analysis, associations between low serum 25-hydroxyvitamin D levels and increased COVID-19 infection rates and worse outcomes were shown [16,17,18], as well as associations between low serum 25-hydroxyvitamin D levels and death [18,19,20,21,22,23,24,25]. On the other hand, other studies found no association between serum 25-hydroxyvitamin D levels and COVID-19 infection [19,26,27]. Dancer et al. (2015) also suggested that a lack of vitamin D contributes to ARDS development because patients with ARDS more often showed a vitamin D deficiency [28]. As ARDS is common in patients with COVID-19, there may be a link between the serum 25-hydroxyvitamin D level and the development of worse outcomes, and several randomised controlled trials have been initiated to investigate this further [29,30,31,32]. However, to date, study results remain inconclusive.

With the current study, we therefore wanted to test retrospectively the hypothesis that vitamin D deficiency is more often associated with a severe outcome in patients hospitalised with COVID-19.

## 2. Materials and Methods

### 2.1. Study Design and Setting

This cross-sectional retrospective observational study was performed at the Cantonal Hospital Baselland (dt. Kantonsspital Baselland (KSBL)), a tertiary teaching hospital in Switzerland located on two sites (Liestal and Bruderholz).

### 2.2. Study Outcomes

The primary study outcome was COVID-19 severity during the course of hospitalisation according to vitamin D status. COVID-19 severity was categorized according to the WHO Ordinal Scale for Clinical Improvement (OSCI) in an adapted version as follows (Table 1):

Secondary exploratory outcomes included the need for oxygen supplementation, in-hospital death, 30 days all-cause rehospitalisation, length of hospital stay alive (LOS), vitamin D supplementation as well as other medications administered, COVID-19-related complications, such as bacterial superinfection, acute kidney failure, gastrointestinal, cardiac and neurologic complications, thromboembolic complications, ventilator-associated pneumonia and multi-organ failure. The COVID-19 wave was also noted and defined as follows (Table 2):

### 2.3. Patient Population: Inclusion and Exclusion Criteria

All adult patients who were hospitalised in the Cantonal Hospital Baselland due to symptomatic, laboratory-confirmed COVID-19 disease with a current available 25-hydroxyvitamin D level (within 30 days before or after hospital admission) between January 2020 and December 2021 were included in the study. COVID-19 needed to be the main diagnosis, which we defined as hospitalisation for COVID-19 or its complications. Complications had to occur within 14 days after testing positive for SARS-CoV-2 (polymerase chain reaction or antigen tests) (see Figure 1). Patients who had declined general consent for the use of health-related data and samples for research purposes were excluded.

### 2.4. Data Collection Process

Patients were selected according to the above-defined criteria. In a case where there was multiple COVID-19 hospitalisation of a particular patient recorded in the defined time period, only the initial COVID-19 hospitalisation of that patient was considered and included in order to avoid bias. Patients’ medical data were collected manually from the in-house electronic patient records and entered into the Research electronic data capture database (REDCap^®^) [34,35].

### 2.5. Defining Vitamin D Deficiency

Serum 25-hydroxyvitamin D levels were measured by liquid chromatography/tandem mass spectrometry (MassChrom^®^ assay from Chromsystems) and were expressed in nmol/L [36]. The serum 25-hydroxyvitamin D level was displayed as a continuous variable as well as a categorical variable. Vitamin deficiency was defined as 25-hydroxyvitamin D level: <50 nmol/L (<20 ng/mL) [37].

Vitamin D supplementation was defined as any dose of vitamin D taken during or after hospitalisation with the intention to increase the levels of 25-hydroxyvitamin D in the body [38].

### 2.6. Statistical Analysis

All statistical analyses were conducted using R statistical software, version 3.4.3 (R Foundation for Statistical Computing) or Statistical Package for the Social Sciences software (SPSS Statistics), Version 24 (IBM).

Categorical variables were shown in absolute and relative frequencies. Continuous variables were shown as mean +/− standard deviation (SD) when normally distributed, or median and interquartile range (IQR) when not normally distributed. Normal distribution was verified using histograms and Q–Q plots.

Categorical variables were analysed with the chi-squared test or Fisher’s exact test. To compare categories of COVID-19 severity and the LOS, which were normally distributed, we used the one-way analysis of variance (ANOVA). For the remaining continuous variables (not normally distributed), we used the Kruskal–Wallis test. For hypothesis testing, we considered *p*-values < 0.05 as statistically significant.

A multivariable logistic regression was performed to assess the relationship between 25-hydroxyvitamin D levels and COVID-19 severity, the need for oxygen supplementation, death and rehospitalisation. A negative binomial regression was performed to assess the relationship between 25-hydroxyvitamin D and LOS. The results of the regression models were presented as odds ratios (OR) or incident rate ratios (IRR) and 95% confidential intervals (CI), with *p* < 0.05 considered statistically significant. To assess the relationship between 25-hydroxyvitamin D level and COVID-19 severity in the multivariable analyses, we categorised severity in two different ways as follows:

Using the ordinal classification mild, moderate, severe, critical and death, using a dichotomised severity variable, whereby mild = mild or moderate and severe = severe, critical or dead.

### 2.7. Ethical Approval

The study was approved by the Ethics Committee of Northwestern and Central Switzerland (EKNZ, BASEC-Number 2021-02479).

## 3. Results

### 3.1. Patient-Based Baseline Characteristics

The study population included 763 patients hospitalised for COVID-19 with a current 25-hydroxyvitamin D status available (Figure 1). Out of 1480 patients, 763 patients were included, whereas 717 patients were excluded due to several reasons (see Figure 1).

The patients’ baseline characteristics are shown in Table 3, which also provides a comparison between the vitamin D deficient and non-deficient groups. The overall median serum 25-hydroxyvitamin D level was 54 nmol/L (IQR 35.00–76.00), and 47% of the patients were vitamin D deficient. All measurements were carried out within 30 days before or after hospital admission, and over 98% were carried out between 5 days before or during hospitalisation.

### 3.2. Primary Study Outcome: COVID-19 Severity According to Vitamin D Status

In both deficient and non-deficient patients, most individuals had mild to moderate COVID-19 severity (81% vs. 84%, *p* = 0.829) (Table 4). Cases of severe and critical COVID-19 were present in both groups, and there was no difference in their distribution. Median serum 25-hydroxyvitamin D levels were similar in patients with mild, moderate, severe, critical and fatal COVID-19 (Table 5).

### 3.3. Secondary Study Outcomes: COVID-19 Clinical Outcome According to Vitamin D Status

Rates of oxygen supplementation, death, rehospitalisation and LOS also did not differ significantly between vitamin D deficient and non-deficient patients (Table 4 and Table 5). 

COVID-19-related complications were more common in vitamin D deficient patients than in non-deficient patients (89% vs. 81%, *p* = 0.003) (Table 4). The most common complication was bacterial superinfection (78% in vitamin D deficient patients vs. 74% in non-deficient patients, *p* = 0.178), followed by acute kidney failure (21% in both groups, *p* = 1.0). Gastrointestinal complications were significantly more common in the vitamin D deficient group (26% vs. 18%, *p* = 0.005) (Figure 2).

Vitamin D supplementation was provided in 37% (Table 6). The supplementation dose of vitamin D recommended and used for most of the deficient patients for correction was 800 IU oral vitamin D3 per day. Fifteen percent of patients were supplemented already before hospitalisation and 22% were started with vitamin D supplementation during the hospitalisation. Vitamin D supplementation was more commonly provided to patients with vitamin D deficiency than to patients without. Furthermore, vitamin D deficient patients were more often supplemented during hospitalisation than non-deficient patients. This finding suggests that measuring vitamin D had an influence on the prescription of vitamin D supplementation during hospitalisation. Nevertheless, more than half (56%) of the vitamin D deficient patients were not supplemented with vitamin D before or during the hospitalisation (Table 6).

Additional organ support, such as vasoactive drugs, dialysis or extracorporeal membrane oxygenation, was provided in 59 patients (8%) (Table 6). All other therapies received are provided in Table 6. According to internal guidelines of the Cantonal Hospital Baselland, based on the Brigham and Women’s Hospital [39] and National Institute of Health (NIH) guidelines [40], different standard COVID-19 treatments were recommended at different time points of this pandemic (Figure 3).

### 3.4. Multivariable Analysis of Primary and Secondary Study Outcome Measures

After adjusting for covariates age, sex, vitamin D supplementation, smoking status, comorbidities, COVID-19 vaccination status and COVID-19 wave, there was no significant association between 25-hydroxyvitamin D level and COVID-19 severity, oxygen supplementation rate, death, rehospitalisation rate, LOS and complication rate (Figure 4).

## 4. Discussion

In this real-life observational study, we found no association of 25-hydroxyvitamin D levels or vitamin D deficiency with the severity of COVID-19, need for oxygen supplementation, rehospitalisation, LOS and in-hospital mortality after adjusting for covariates age, sex, vitamin D supplementation, smoking status, comorbidities, COVID-19 vaccination status and COVID-19 wave. COVID-19-related complications were more common in the vitamin D deficient group.

Our findings correspond to Hernández et al.’s study conducted in 2021, which showed no association between 25-hydroxyvitamin D levels or vitamin D deficiency and COVID-19 severity [41]. Similarly, our study and Im et al. [42] observed non-statistically significant trends for more vitamin D deficient patients in the critical COVID-19 group compared to the other severity groups. Our study and Georgoulis et al. included similar patients, with more than 80% having at least one comorbidity and one-third being obese. In the vitamin D deficient groups, there were more obese patients than in the non-deficient groups, which corresponds to obesity being a risk factor for vitamin D deficiency [43].

Our findings do not correspond to a systematic review where aged (mean > 60 years) vitamin D deficient patients had greater risk of adverse outcomes compared to vitamin D non-deficient patients [44]. However, the studies analysed in the review only investigated small cohorts ranging from 20 to 185 patients, which may not have been representative for all COVID-19 waves [44]. In our study, no statistically significant association was found between vitamin D levels and in-hospital death; this corresponds to the findings of Kazemi et al. [45]. Our study also found no statistically significant association between 30-day all-cause rehospitalisation and 25-hydroxyvitamin D levels. Our study and Reis et al. found no association of 25-hydroxyvitamin D levels with the LOS. However, in their study, a statistically non-significant trend for longer LOS was observed in patients with vitamin D deficiency, a finding that our study did not corroborate [46]. It is important to note that, in this study, we used the cutoff of 50 nmol/L (20 ng/mL) for defining vitamin D deficiency, in line with the definition provided by Holick et al. [37]. However, levels below 75 nmol/L (30 ng/mL) might already affect certain health risks, including the risk for infection, and the goal should be to maintain the level above this value [47,48,49].

Vitamin D supplementation was provided in 37% (15% were supplemented already before hospitalisation, while 22% started with the supplementation during the hospitalisation). Concerning COVID-19 treatment, the majority of the patients were treated with Remdesivir and antibiotics throughout the COVID-19 period.

Our study included patients whose serum 25-hydroxyvitamin D levels were measured 30 days before or after hospital admission, which might have resulted in vitamin D supplementation prior to infection in certain patients and effectively higher levels at the time of admission. However, the vast majority (98%) had the 25-hydroxyvitamin D assessment up to 5 days before admission or during admission, indicating that the measurements were most often taken during acute infection. Furthermore, only 21 of the vitamin D deficient patients (6%) had received vitamin D supplementation before admission. Vitamin D supplementation was evenly distributed between the two groups, but vitamin D deficient patients were less often supplemented with vitamin D before the hospitalisation but more often started supplementation during the hospitalisation. However, we did not consider the exact dosage, which might have provided a better understanding of the role of supplementation.

More than 80% of our patients suffered from a COVID-19-related complication, mostly bacterial superinfection. There were significantly more vitamin D deficient patients suffering from a complication compared to the non-deficient patients. Previous research has led to the assumption that a non-deficient vitamin D status could play an immunoprotective role in respiratory infections by promoting the recruitment of immune cells to the infection site and the induction of infected cell apoptosis, leading to clearance of respiratory pathogens [50,51,52]. Furthermore, vitamin D has been shown to reduce inflammatory activity by suppressing cytokine production, which may lead to fewer thromboembolic complications due to a less procoagulatory setting [53]. However, the model using continuous 25-hydroxyvitamin D values as a potential predictor for complications did not confirm this association. When looking at the complications in detail, a significant difference between vitamin D deficient and non-deficient patients was found in gastrointestinal complications. This finding adds to the other research, suggesting that vitamin D may play a role in intestinal health as it might alter the gut immune function, predisposing deficient patients to gastrointestinal complications [54,55,56]. COVID-19, on the other hand, is known to be associated with gastrointestinal complications, such as ulcerative colitis and acute pancreatitis [57,58].

### Strengths and Limitations

In this study, we included COVID-19 confirmed hospitalised cases over 2 years of the pandemic, enabling us to study a large and heterogeneous COVID-19 cohort. In addition to categories of vitamin D deficiency/non-deficiency, we assessed continuous 25-hydroxyvitamin D values as a potential predictor for different outcomes. Models were adjusted for covariates, minimising any bias effect, resulting in robust models. We also used an established classification for disease severity with the OSCI, enabling us to compare our findings with other studies in the field. Our study only included patients with an available 25-hydroxyvitamin D measurement, which might have resulted in a selection bias and limited comprehensive understanding of all hospitalised COVID-19 patients. Variables such as vital signs for more detailed COVID-19 severity definition, osteoporosis and hyperparathyroidism as comorbidities and risk factors in vitamin D deficiency were not assessed. As a retrospective study, one potential limitation is that patients were included if they had a 25-hydroxyvitamin D level taken before, during or after COVID-19. This could introduce confounding elements to the study design as 25-hydroxyvitamin D levels are likely to fluctuate throughout the disease and may be influenced by various factors, such as disease severity, medications and comorbidities. Additionally, the timing of serum 25-hydroxyvitamin D level measurement may vary between the included patients (in our case, measured in 98% of the patients before or during admission), which could affect the comparability of the data collected. Also, vitamin D supplementation was investigated in a general manner without considering the exact dosage.

## 5. Conclusions

In this retrospective study, we found no association of vitamin D deficiency or 25-hydroxyvitamin D levels with COVID-19 severity, oxygen supplementation, mortality, rehospitalisation or LOS after adjusting for covariates. Vitamin D deficient patients seem to be more at risk for COVID-19-related complications, especially gastrointestinal ones. Since we analysed data from a large and heterogeneous cohort over the course of two years of the pandemic, these results present an important contribution to the controversially discussed role of vitamin D in COVID-19.

## Figures and Tables

**Figure 1 jcm-12-05520-f001:**
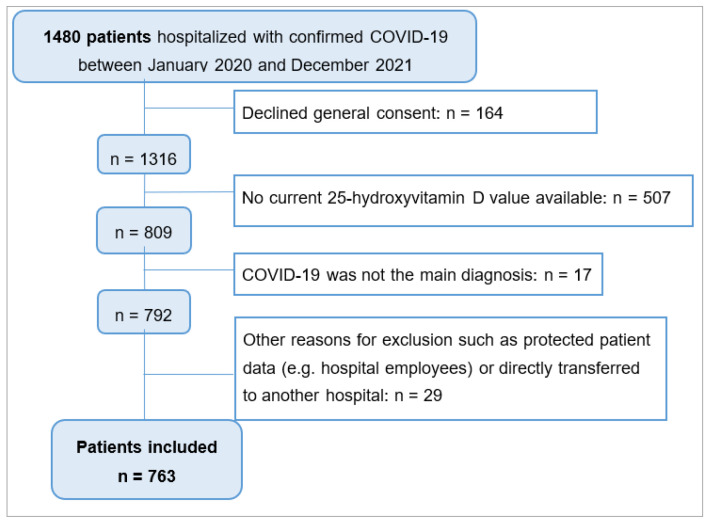
Flowchart diagram for patient selection process. Fields with a white background: number of excluded patients due to this reason for exclusion.

**Figure 2 jcm-12-05520-f002:**
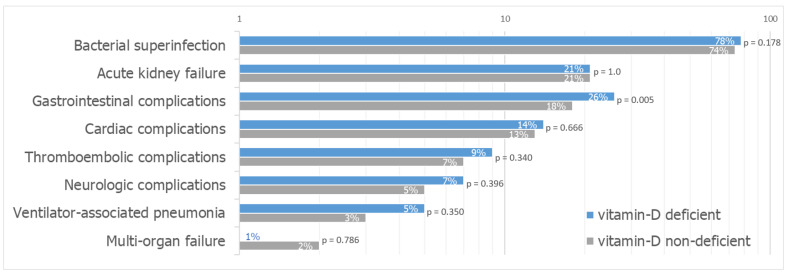
COVID-19-related complications in vitamin D deficient vs. non-deficient patients (shown in % per category, logarithmic scale).

**Figure 3 jcm-12-05520-f003:**
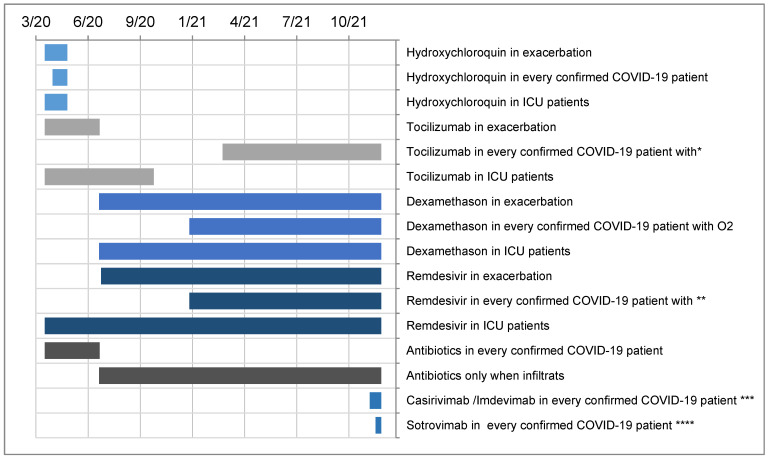
Overview timeline of COVID-19 therapies adapted according to the KSBL guidelines. * Dexamethasone and aggravation of O_2_ need within 3 days or need of NIV/HFNO/intubation. ** O_2_ and within the first 8 days since the beginning of symptoms. *** within 10 days since the beginning of symptoms and negative SARS-CoV-2 N and S antibody or only S after vaccination. **** When Omicron was dominant, replaced Casirivimab/Imdevimab.

**Figure 4 jcm-12-05520-f004:**
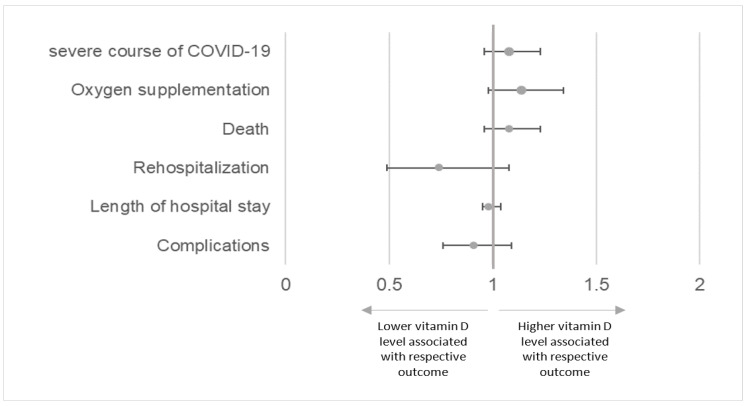
Odds ratio (or incident rate ratio ^1^) with 95% confidence intervals for different outcomes depending on serum vitamin D level (per 25 nmol/L increase), adjusted for age, sex, vitamin D supplementation, smoking status, comorbidities, vaccination status, and COVID-19 wave. Severe course of COVID-19 was defined as WHO category severe, critical or death. ^1^ Incident rate ratio for outcome length of hospital stay.

**Table 1 jcm-12-05520-t001:** Definition of COVID-19 severity.

Mild	Hospitalised without oxygen therapy
Moderate	Hospitalised with oxygen by a nasal prong or mask
Severe	Hospitalised with oxygen by non-invasive ventilation (NIV) or high-flow nasal oxygen (HFNO)
Critical	Hospitalised and intubated, incl. patients with additional organ support (dialysis, extracorporeal membrane oxygenation (ECMO), vasoactive drugs)
Dead	

Modified from the WHO Ordinal Scale for Clinical Improvement [33].

**Table 2 jcm-12-05520-t002:** Definition of COVID-19 waves.

First wave	27 February 2020 to 30 June 2020
Second wave	1 July 2020 to 28 February 2021
Third wave	1 March 2021 to 30 June 2021
Fourth wave	1 July 2021 to 15 October 2022
Fifth wave	16 October 2021 to 31 December 2021

**Table 3 jcm-12-05520-t003:** Patient baseline characteristics by vitamin D status.

	Total Study Population	Vitamin D Deficient (<50 nmol/L)	Vitamin D Non-Deficient (≥50 nmol/L)	*p*-Value
n (%)	763 (100)	343 (47)	420 (53)	
Age (years), mean ± SD	65.28 ± 16.36	64.09 ± 17.27	66.26 ± 15.53	0.068
Sex female, n (%)	329 (43.1)	121 (35.3)	208 (49.5)	<0.001 ***
BMI (kg/m^2^) mean ± SD	28.18 ± 5.45	28.95 ± 5.72	27.56 ± 5.14	0.002 **
Smoking status				0.409
Non-smoker, n (%)	107 (14.0)	50 (14.6)	57 (13.6)	
Smoker, n (%)	28 (3.7)	11 (3.2)	17 (4.0)	
Former smoker, n (%)	98 (12.8)	51 (14.9)	47 (11.2)	
Unknown, n (%)	530 (69.5)	231 (67.3)	299 (71.2)	
Pack years, median (IQR)	40 (23.5–52.3)	40 (25–52.5)	37.5 (20.8–52.3)	0.504 ^‡^
Patients with comorbidity, n (%)	642 (84.1)	295 (86)	347 (82.6)	0.240
Number of comorbidities, median (IQR)	3 (2–4)	3 (2–4)	3 (2–4)	0.212 ^‡^
Arterial hypertension, n (%)	360 (47.2)	160 (46.6)	200 (47.6)	0.846
Cardiac diseases, n (%)	231 (30.3)	112 (32.7)	119 (28.3)	0.225
Obesity, n (%)	225 (29.7)	116 (34.2)	109 (26.1)	0.018 *
Diabetes, n (%)	166 (21.8)	83 (24.2)	83 (19.8)	0.165
Chronic respiratory diseases, n (%)	172 (22.5)	83 (24.2)	89 (21.2)	0.367
Immunosuppression, n (%)	60 (7.9)	18 (5.2)	42 (10)	0.022 *
Chronic kidney diseases, n (%)	161 (21.1)	74 (21.6)	87 (20.7)	0.825
Thyroid diseases, n (%)	93 (12.2)	33 (9.6)	60 (14.3)	0.065
Malignant diseases, n (%)	100 (13.1)	40 (11.7)	60 (14.3)	0.337
Active Cancer, n (%)	32 (4.2)	11 (3.2)	21 (5)	0.295
Chronic liver diseases, n (%)	58 (7.6)	27 (7.9)	31 (7.4)	0.907
Neurological disorders, n (%)	196 (25.7)	73 (21.3)	123 (29.3)	0.015 *
Electrolyte imbalance, n (%)	181 (23.7)	75 (21.9)	106 (25.2)	0.315
COVID-19 vaccination status	661 (87)			0.156
Negative (unvaccinated), n (%)	616 (81)	262 (76)	354 (84)	
Positive (vaccinated), n (%)	45 (6.8)	23 (8.1)	22 (5.9)	
Time between vaccination and admission (days), median (IQR)	63 (12.5–200.5)	95.5 (10–214.5)	46 (13–183)	0.978 ^‡^
25-hydroxy vitamin D measured 5 days before admission or during admission, n (%), mean	751 (98.4)	339 (98.8)	412 (98.1)	0.601

BMI = body mass index (kg/m^2^), SD = standard deviation, IQR = interquartile range. Number of comorbidities summed the presence of the following comorbidities: arterial hypertension, cardiac diseases, obesity (BMI ≥ 30 kg/m^2^), diabetes, chronic respiratory disease, immunosuppression, chronic kidney disease, thyroid disease, malignant disease, chronic liver disease, neurological disorder and electrolyte imbalance. Continuous variables are shown as mean ± SD and a *t*-test was applied when normally distributed, or as median with IQR and a Mann–Whitney U test (^‡^) was applied when not normally distributed. Categorial variables are shown as absolute and relative frequencies and a chi-squared test was used for the analyses. *p*-values < 0.05 were considered statistically significant. Significance codes: * <0.05, ** <0.01, *** <0.001.

**Table 4 jcm-12-05520-t004:** COVID-19 severity and outcome by vitamin D status.

	Total	Vitamin D Deficient (<50 nmol/L)	Vitamin D Non-Deficient (≥50 nmol/L)	*p*-Value
n (%)	763 (100)	343 (45)	420 (55)	
COVID-19 severity category	763 (100)			0.561 ^a^
Mild, n (%)	255 (33.4)	112 (32.7)	143 (34)	
Moderate, n (%)	374 (49)	166 (48.4)	208 (49.5)	
Severe, n (%)	27 (3.5)	11 (3.2)	16 (3.8)	
Critical, n (%)	38 (5)	22 (6.4)	16 (3.8)	
Death, n (%)	69 (9)	32 (9.3)	37 (8.8)	
Oxygen supplementation, n (%)	508 (66.6)	231 (67.3)	277 (66)	0.742 ^a^
Death, n (%)	69 (9)	32 (9.3)	37 (8.8)	0.903 ^a^
COVID-19-related complications	647 (84.8)	306 (89.2)	341 (81.2)	0.003 ^a^
Length of hospital stay alive (LOS) (days), median (IQR)	7 (4–11)	7 (4–11)	7 (4–11)	0.865 ^b^
Rehospitalised at KSBL within 30 days, n (%)	32 (4.2)	17 (5)	15 (3.6)	0.443 ^a^

LOS = length of hospital stay (days), SD = standard deviation, IQR = interquartile range. ^a^ Chi-squared test. ^b^ Mann–Whitney U test. Most of the cases had a mild to moderate COVID-19 course throughout all the waves, and oxygen supplementation was used more or less the same over the COVID-19 waves.

**Table 5 jcm-12-05520-t005:** Median serum vitamin D values for population sub-groups.

	Serum Vitamin D (nmol/L), Median (IQR)	*p*-Value
COVID-19 severity		
Mild	55 (36–76)	
Moderate	54 (36–76)	
Severe	55 (36–73)	0.649 ^₤^
Critical	45 (34–71)	
Dead	51 (29–82)	
Oxygen supplementation		
Yes	53 (36–76)	0.837 ^‡^
No	55 (35–76)	
Death		
Yes	51 (29–82)	0.836 ^‡^
No	54 (35–74)	
Rehospitalisation at KSBL within 30 days		
Yes	45 (32–73)	0.402 ^‡^
No	54 (36–76)	
Length of hospital stay		
1–3 days	52 (38–79)	0.942 ^₤^
4–7 days	56 (37–74)	
8–13 days	52 (34–71)	
14–21 days	55 (36–78)	
22–28 days	52 (36–86)	
>28 days	50 (38–70)	
COVID-19-related complications		
Yes	52 (43–82)	0.016 ^‡^*
No	64 (35–74)	

IQR = interquartile range. Continuous variables are shown as mean ± SD when normally distributed or as median with IQR when not normally distributed, and a Mann–Whitney U test (^‡^) was applied for analyses between two groups or a Kruskal–Wallis test (^₤^) between several groups. *p*-values < 0.05 were considered statistically significant. Significant codes: * <0.05.

**Table 6 jcm-12-05520-t006:** Analysis of COVID-19 therapies according to vitamin D status.

	Total	Vitamin D Deficient (<50 nmol/L)	Vitamin D Non-Deficient (≥50 nmol/L)
n (%)	763 (100)	343 (45)	420 (55)
Antibiotics, n (%)	592 (78)	264 (77)	328 (78)
Specific COVID-19 therapy, n (%)	513 (67)	236 (69)	277 (66)
Anticoagulants and platelet aggregation inhibitors, n (%)	750 (98)	336 (98)	414 (98)
Antihypertensive therapy, n (%)	407 (53)	167 (49)	240 (57)
Inhalation therapy, n (%)	239 (31)	102 (30)	137 (33)
Vitamin D supplementation, n (%)	279 (37)	150 (44)	129 (31)
Only before hospitalization, n (%)	111 (15)	21 (14)	90 (70)
Only during hospitalization, n (%)	168 (22)	129 (86)	39 (30)
No vitamin D supplementation, n (%)	484 (63)	193 (56)	291 (69)
Additional Organ Support	59 (8)	27 (8)	32 (8)
Dialysis, n (%)	5 (1)	1 (4)	4 (1)
Vasoactive drugs, n (%)	45 (6)	21 (6)	24 (6)
ECMO, n (%)	9 (1)	5 (19)	4 (1)

Vitamin D deficient patients had less antihypertensive therapy (supplemented: 49% vs. non-supplemented: 57%) and less inhalation therapy (supplemented: 30% vs. non-supplemented: 33%). Antibiotics were provided in 78%, whilst specific COVID-19 therapies were only provided in 67%, with dexamethasone being the most frequent.

## Data Availability

The data presented in this study are available on request from the corresponding author. The data are not publicly available due to data privacy restrictions.

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
