# Peer review of "Is Serum 25-Hydroxyvitamin D Level Associated with Severity of COVID-19? A Retrospective Study"

_jcm, 2023, doi:10.3390/jcm12175520_

Round 1
Reviewer 1 Report
1. I would not rely much on 25-OH-D levels if they were performed (almost) 1 month prior to hospitalization due to COVID-19. It is possible that certain patients, in case low vitamin D was detected, might have started taking high vitamin D doses (included OTC supplements); therefore, the recent 25-OH-D levels (during the COVID-19) might have been significantly higher than before hospitalization and this might influence the associations (i. d., the result of statistical analysis would be quite different). I suggest the perform again the analysis including only those patients that had 25-OH-D data from the labs performed at the hospital or not more that 1 week prior to hospitalization.
2. The cut-off 20 ng/ml is not very appropriate in this kind of trials. Patients having 25-OH-D levels between 20 and 30 ng/ml are vitamin D insufficient and might also experience various health problems, e.g., higher risk for certain infectious diseases, as those having levels below 25-OH-D. In other words, if the authors presume that patients with 25-OH-D levels 20 - <30 ng/ml are “well” regarding vitamin D status and could be put into the same group as those having >30 ng/ml – it is wrong! I suggest redoing the statistical analysis using the cut-off 30 ng/ml.
3. Line 52. “25-hydroxyvitamin D <…> is the only vitamin D metabolite that is used to determine whether a patient is vitamin D deficient, sufficient or intoxicated.”. Regarding vit. D intoxication, it is totally wrong. Very high levels of 25-OH-D mean that the patient has overdosed vitamin D, but overdosing is not equal to toxicity. Many patients with very high levels of 25-OH-D are not intoxicated. Intoxication with vitamin D is based on clinically significant hypercalcemia, usually accompanied by hypercalciuria and, in case of long-lasting hypercalcemia – calcification of different organs and tissues.
4. Figure 1. The textbox with text “No current 25-hydroxyvitamin D value” should contain the number of those patients. I think “n = 507” should be added.
The same remark regarding textbox “Other reasons for exclusion such as protected patient data (e.g. hospital employees) or directly transferred” – “n = 29” could be added.
None
Author Response
Response to Reviewer 1 Comments
Thank you very much for your very valuable and constructive comments. We hope our revised version of the manuscript can settle your previous concerns. Following, you find a point-by-point response to the stated comments.
Point 1: I would not rely much on 25-OH-D levels if they were performed (almost) 1 month prior to hospitalization due to COVID-19. It is possible that certain patients, in case low vitamin D was detected, might have started taking high vitamin D doses (included OTC supplements); therefore, the recent 25-OH-D levels (during the COVID-19) might have been significantly higher than before hospitalization and this might influence the associations (i. d., the result of statistical analysis would be quite different). I suggest the perform again the analysis including only those patients that had 25-OH-D data from the labs performed at the hospital or not more that 1 week prior to hospitalization.
Response 1: We thank the reviewer for this very valuable comment. However, since 98% of the included patients had their 25-OH-D level measured up to 5 days before admission (as stated in the baseline characteristics table). Furthermore, only 21 of vitamin D deficient patients (6%) had received a vitamin D supplementation before admission. Thus, we believe that potential supplementation after the measurement in individual patients would not influence our findings. We added this important line of thought to the discussion section.
Point 2: The cut-off 20 ng/ml is not very appropriate in this kind of trials. Patients having 25-OH-D levels between 20 and 30 ng/ml are vitamin D insufficient and might also experience various health problems, e.g., higher risk for certain infectious diseases, as those having levels below 25-OH-D. In other words, if the authors presume that patients with 25-OH-D levels 20 - <30 ng/ml are “well” regarding vitamin D status and could be put into the same group as those having >30 ng/ml – it is wrong! I suggest redoing the statistical analysis using the cut-off 30 ng/ml.
Response 2: Thank you for pointing out this important circumstance. We used the cut-off of 20ng/ml for vitamin D deficiency as recommended by different societies/guidelines and expert bodies, such as the clinical guidelines of Endocrine Society Task Force on vitamin D and the Institute of Medicine. This cut-off makes our results comparable to previous research in the field. Being aware of the fact that a cut-off might not reveal all underlying associations, we also assessed continuous 25-OH-D levels as a potential predictor in logistic and negative binomial regression models (Figure 4). We dedicated a passage in the discussion section to this concern.
Point 3: Line 52. “25-hydroxyvitamin D <…> is the only vitamin D metabolite that is used to determine whether a patient is vitamin D deficient, sufficient or intoxicated.”. Regarding vit. D intoxication, it is totally wrong. Very high levels of 25-OH-D mean that the patient has overdosed vitamin D, but overdosing is not equal to toxicity. Many patients with very high levels of 25-OH-D are not intoxicated. Intoxication with vitamin D is based on clinically significant hypercalcemia, usually accompanied by hypercalciuria and, in case of long-lasting hypercalcemia – calcification of different organs and tissues.
Response 3: Thank you for pointing out this mistake. We have taken note of that and adjusted the passage accordingly.
Point 4: Figure 1. The textbox with text “No current 25-hydroxyvitamin D value” should contain the number of those patients. I think “n = 507” should be added.
The same remark regarding textbox “Other reasons for exclusion such as protected patient data (e.g. hospital employees) or directly transferred” – “n = 29” could be added.
Response 4: Thank you for pointing this out. We adjusted Figure 1 accordingly.
Reviewer 2 Report
This is a hospital based cross-sectional, retrospective, observational study regarding the association between vitamin D status and oxygen supplementation, death, COVID-19 related complications, length of hospital stay alive (LOS) (days), rehospitalised at KSBL within 30 days, COVID-19 therapies after adjusting for age, sex, vitamin D supplementation, smoking status, comorbidities, COVID vaccination status, and COVID wave. The idea is good and the valuable for publication. But there are some points to address before final decision.
1. Introduction
-It is not necessary to include subsections in the “Introduction” section.
2. Materials and Methods
-The structure of writing applied in the manuscript is not in concordance with the journal format. For instance, in the Method sections authors used bullet points which is not custom in the writing of the scientific papers.
- As the dada collection has been occurred in the second wave of COVID-19, CT scan assay had been a routine test for the diagnosis of this disease. Why hasn’t this test considered as a criterion for including inpatients in the study? Serologic tests are not reliable tests for the COVID-19 diagnosis.
-“Data collection process” section needs to be grammatically revised. It is so incomprehensible in this way.
-The dosage of vitamin D supplements received by each patient have not been considered in this study. As the amount of vitamin D taken by patients has a powerful effect on their serum levels of 25 (OH)D3, how do you verify that all patients have taken the same dose? How do you control the obtained results in term of dose of vitamin D supplements and dietary vitamin D intake? Especially that you have not inserted a reference for sentence “Vitamin D supplementation was defined as any dose of vitamin D taken during or after hospitalization” on page 5.
3. Results
On page 9, the sentences “COVID-19 related complications were more common in vitamin D deficient patients than in non-deficient 188 patients (89% vs. 81%, p = 0.003) (Table 2). The most common complication was bacterial superinfection (78% 189 in vitamin D deficient patients, vs. 74% in non-deficient patients), followed by acute kidney failure (21% in 190 both groups), gastrointestinal (26% vs 18%) and cardiac complications (14% vs. 13%) (Figure 2).”, P-values for the differences between two groups has not been included in some cases. Please include them in Figure 2 and the text.
4. Discussion
-There are quite a few grammatical points in this section that should be revised.
-the most important result of this study is that there is a significant association between vitamin D levels and complications related to the COVId-19. But there is not adequate explanations and discussion about this results in “Discussion” section.
- The authors pointed out their obtained results and the results of other studies. But, it is necessary to explain their results more with some probable mechanisms.
- it is not needed to put “Study strengths and Study limitations” in separated subsection. Please remove the subsections.

Author Response
Response to Reviewer 2 Comments
Thank you very much for your very valuable and constructive comments. We hope our revised version of the manuscript can settle your previous concerns. Following, you find a point-by-point response to the stated comments.
Introduction
Point 1: It is not necessary to include subsections in the “Introduction” section.
Response 1: Thank you for pointing this out. We have removed the subsections.
Materials and Methods
Point 2: The structure of writing applied in the manuscript is not in concordance with the journal format. For instance, in the Method sections authors used bullet points which is not custom in the writing of the scientific papers.
Response 2: We thank the reviewer for this comment. We have removed the bullet points and reformatted the relevant sections into tables where it was important for readability.
Point 3: As the dada collection has been occurred in the second wave of COVID-19, CT scan assay had been a routine test for the diagnosis of this disease. Why hasn’t this test considered as a criterion for including inpatients in the study? Serologic tests are not reliable tests for the COVID-19 diagnosis.
Response 3: Thank you very much for this interesting question. While CT scan assays were routinely used during the second wave of COVID-19, they were not used as an inclusion criterion for various reasons. They do not provide a concrete diagnosis of COVID-19 but focus on the involvement of the lungs. Furthermore, due to the rapid change of diagnostic standards in COVID-19 during the pandemic, we solely used the ICD-10 coding preformed in our coding department as the main inclusion criteria in order to have a consistent criterion throughout all waves. We trust that the coding department was implementing all new research into the coding as soon as available and therefore believe that this is the best way of including all COVID-19 patients without missing anyone.
Serological testing, on the other hand, was used for the definition of the time frame for relevant complications. We rephrased the paragraph in order to prevent misunderstanding regarding serological testing.
Point 4: “Data collection process” section needs to be grammatically revised. It is so incomprehensible in this way.
Response 4: Thank you for pointing this out. We revised the paragraph for the sake of comprehensibility and deleted irrelevant passages.
Point 5: The dosage of vitamin D supplements received by each patient have not been considered in this study. As the amount of vitamin D taken by patients has a powerful effect on their serum levels of 25 (OH)D3, how do you verify that all patients have taken the same dose? How do you control the obtained results in term of dose of vitamin D supplements and dietary vitamin D intake? Especially that you have not inserted a reference for sentence “Vitamin D supplementation was defined as any dose of vitamin D taken during or after hospitalization” on page 5.
Response 5: Thank you for this important comment. While it is true that the amount of vitamin D taken by patients has an effect on 25 (OH) D3, this study did not directly consider the exact dosage taken. Instead, we focused on whether or not vitamin D was supplemented during or after hospitalization, regardless of the dosage. We acknowledge this as a limitation of our study, as different doses can lead to different outcomes; we added this information to the discussion and limitations section. In future studies, more data on dosage and duration of vitamin D supplementation could be collected to provide more understanding. As for controlling for dietary vitamin D intake, we also consider it a limitation in a study of this nature and mentioned it in the limitations setion.
Results
Point 6: On page 9, the sentences “COVID-19 related complications were more common in vitamin D deficient patients than in non-deficient 188 patients (89% vs. 81%, p = 0.003) (Table 2). The most common complication was bacterial superinfection (78% 189 in vitamin D deficient patients, vs. 74% in non-deficient patients), followed by acute kidney failure (21% in 190 both groups), gastrointestinal (26% vs 18%) and cardiac complications (14% vs. 13%) (Figure 2).”, P-values for the differences between two groups has not been included in some cases. Please include them in Figure 2 and the text.
Response 6:
We thank the reviewer for this important notion. We included the p-values into Figure 2 and in the text passage, where relevant. We also highlighted the significant difference between vitamin D deficient and non-deficient regarding gastrointestinal complications in the text.
Discussion
Point 7: There are quite a few grammatical points in this section that should be revised.
Response 7: Thank you for pointing this out. We corrected grammatical errors that were present in this section.
Point 8: The most important result of this study is that there is a significant association between vitamin D levels and complications related to the COVId-19. But there is not adequate explanations and discussion about this results in “Discussion” section.
Response 8: Thank you very much for this highly relevant comment. We have added more explanation to the discussion section regarding this result.
Point 9: The authors pointed out their obtained results and the results of other studies. But, it is necessary to explain their results more with some probable mechanisms.
Response 9: We thank the reviewer for this comment. We have added probable mechanisms to the discussion of the results regarding COVID-19 related complications. Furthermore, we tried to find probable reasons for the deviation from results of previous studies.
Point 10: it is not needed to put “Study strengths and Study limitations” in separated subsection. Please remove the subsections.
Response 10: Thank you for pointing this out. We adapted the section accordingly.
Round 2
Reviewer 1 Report
No comments.
Author Response
Thank you very much for the careful review of our manuscript and the positive feedback.
Reviewer 2 Report
Thank you for addressing the comments. Some points have remained to consider as follows
1. There is no need for punctuation at the title's end.
2. The figure 1 does not have any caption.
Good luck!
Author Response
Thank you once again for your careful review of our manuscript and the overall positive feedback.
In the revised version of our manuscript we addressed your recent comments as follows and hope you will now find our manuscript suitable for publication:
Point 1: There is no need for punctuation at the title's end
Response 1: Thank you for pointing this out, we have removed the punctuation.
Point 2: The figure 1 does not have any caption.
Response 2: The caption of Figure 1 is included in the manuscript on page 5, line 94-95. We have highlighted it for your attention. The caption is not included in the image file, as per author instructions.